# Learning Environment-Aware Affordance for 3D Articulated Object Manipulation under Occlusions

**Ruihai Wu**[1,4]*    **Kai Cheng**[2]*
**Yan Shen**[1,4]    **Chuanruo Ning** [2]    **Guanqi Zhan** [3]    **Hao Dong** [1,4]†
[1]CFCS, School of CS, PKU    [2]School of EECS, PKU    [3]University of Oxford
[4]National Key Laboratory for Multimedia Information Processing, School of CS, PKU

## Abstract

Perceiving and manipulating 3D articulated objects in diverse environments is essential for home-assistant robots. Recent studies have shown that point-level affordance provides actionable priors for downstream manipulation tasks. However, existing works primarily focus on single-object scenarios with homogeneous agents, overlooking the realistic constraints imposed by the environment and the agent's morphology, *e.g.*, occlusions and physical limitations. In this paper, we propose an environment-aware affordance framework that incorporates both object-level actionable priors and environment constraints. Unlike object-centric affordance approaches, learning environment-aware affordance faces the challenge of combinatorial explosion due to the complexity of various occlusions, characterized by their quantities, geometries, positions and poses. To address this and enhance data efficiency, we introduce a novel contrastive affordance learning framework capable of training on scenes containing a single occluder and generalizing to scenes with complex occluder combinations. Experiments demonstrate the effectiveness of our proposed approach in learning affordance considering environment constraints.

## 1   Introduction

Articulated objects, such as doors and drawers, exist everywhere in our daily life. Perceiving and manipulating these objects present crucial yet challenging tasks in computer vision and robotics. Unlike rigid objects, articulated objects exhibit diverse articulation types and functionally important articulated parts crucial for human and robot interactions. Numerous research endeavors have been investigating articulated objects broadly, encompassing joint parameters estimation [41, 47], part pose estimation [22, 23], kinematic structure estimation [34, 33], digital twins generalization [13, 10], articulated part robotic manipulation [25, 44, 46, 4] and few-shot policy adaptation [42].

However, most existing works for manipulating articulated objects primarily focus on single-object scenarios with homogeneous agents, such as flying grippers [46, 25, 44] or fixed-position robot arms [6]. Consequently, these approaches tend to develop **object-centric** representations and policies, neglecting the realistic constraints imposed by both the environment and the agent's morphology. These constraints are commonplace in real-world scenarios and their oversight limits the applicability and performance of the manipulation tasks. For example, successfully opening a cabinet door that is obstructed by occluders not only depends on the properties of the target door but also heavily relies on the robot's position and the way it interacts (*e.g.*, colliding or bypassing) with the occluders.

We take a significant step towards manipulating articulated objects in a more realistic setting, *i.e.*, considering constraints imposed by the environment and robot. Such a task encounters the combi-

---

*Equal contribution. Author ordering determined by coin flip.
†Corresponding author.

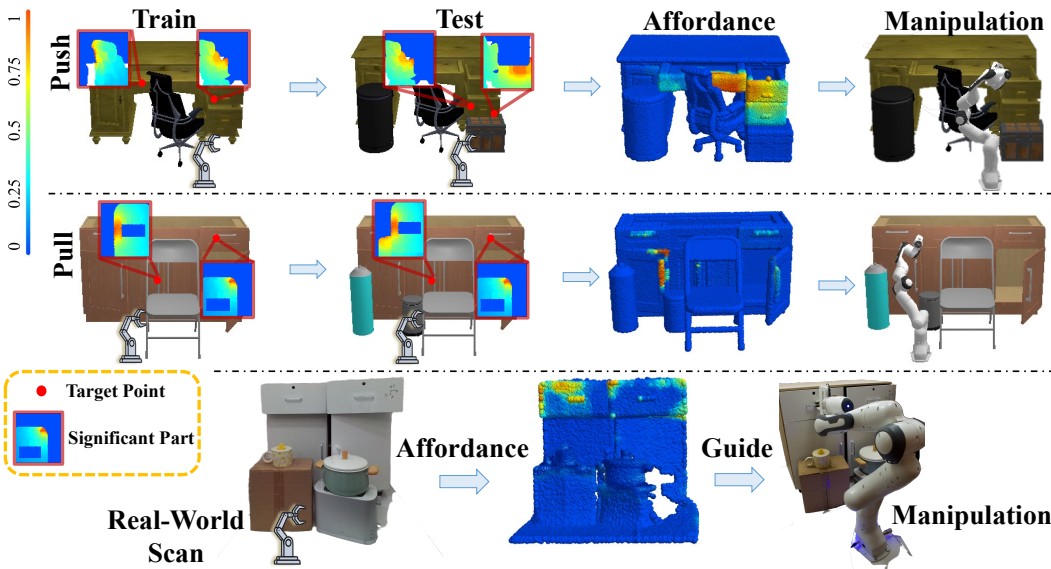

Figure 1: **Learning Environment-Aware Affordance for Articulated Object Manipulation.** As the complexity of occluder combinations grows exponentially, we leverage the property that target manipulation point (**Red Point**) conditioned *significant parts of occluders* that impact the manipulation usually have limited local areas (**Red Point's** corresponding **Red Box**) even in complex scenes. Aware of such significant parts, our model can be trained on one-occluder scenes (**Train**) and generalize to multiple occluder combinations (**Test**, **Affordance**). The learned affordance provides actionable information for articulated object manipulation (**Manipulation**). Our model predicts reasonable affordance on real-world scanned point clouds (**Real-world Scan**).

natorial explosion challenge in complexity. To be specific, the substantial variability of occluders, characterized by their numbers, geometries, positions, and poses, leads to exponential complexity in scene data distribution [49, 18]. This introduces a significant obstacle to training a model that can comprehensively understand the diversity and intricacies of different scenes within the limits of available data. Moreover, sampling-based motion planning [19, 35], a kind of commonly used approach for planning and manipulation, also faces performance degradation with increasing occluders [38] as the probability of valid sampling rapidly diminishes.

We propose the use of **point-level** representations to tackle the challenge outlined above. Specifically, we exploit a notable property: even in situations where the combinations of occluders are complex, given the target manipulation point and the robot, the occluder parts that significantly affect the manipulation are typically confined to a limited area. This is clearly illustrated in Figure 1 (Row 1, Column 2), where, in the case of pushing a drawer obstructed by multiple occluders, the portions of the occluders that could potentially collide with the robot are confined to a specific area (shown as areas in the two boxes with bright colors in the heatmap). This confinement results from the predetermined position of the robot and the target manipulation point, which limits the range of possible trajectories for the manipulation. Hence, when we contemplate the manipulation of the target object at the **point level**, the complexity of these significant scene components remains manageable, despite the possible addition of unseen occluders.

Further, we take inspiration from **affordance** for robotic manipulation, which provides actionable priors of the target at the point level and thus guides manipulation policies, and has exhibited remarkable efficacy in 3D articulated object manipulation [25, 44, 42, 6] and other manipulation tasks [54, 26, 43]. Unlike these works that focus solely on single objects, we introduce the concept of **environment-aware affordance** for articulated object manipulation, which integrates **object-centric actionable priors** with **environment constraints** at the point level. Specifically, in a scene featuring a target articulated object, diverse occlusion objects, and a robot at different positions, we aim to learn the per-point actionable information of the target object, taking into account the constraints imposed by the environment and the agent. Furthermore, in light of Gibson's theory of affordance

(the originator of affordance theory), where affordance is defined as the different possibilities of action that the **environment** offers to an **agent** [7], our proposed task is not only more realistic but also more closely aligned with this foundational theory compared to preceding works.

The point-level representation we propose above shows the potential to train a *significant-part-aware* model in simple scenes while capable of generalizing to more complex scenes. This approach leverages the aforementioned point-level significant parts, whose complexity remains manageable across a spectrum from simple to complicated scenes. To facilitate learning of significant-part-aware scene representations, we design a robot-target conditioned occlusion field.

Further, we introduce a contrastive learning method that refines our learned representations to selectively disregard insignificant elements while maintaining sensitivity to similar, significant parts across varied scenes. Consequently, our proposed learning framework can be trained on scenes with a single occluder, yet generalizable to scenes with numerous novel occluder combinations. This strategy effectively mitigates the issue of combinatorial explosion in a data-efficient manner.

We conduct experiments using SAPIEN physics simulator [45] equipped with large-scale PartNet-Mobility [27] and ShapeNet [1] datasets. To assess our framework's performance and data efficiency, we initially train it on scenes containing only one occluder, with additional augmented contrastive scenes that incorporate an extra occluder. We then test our framework on significantly more complex scenes, featuring diverse combinations of novel occluders. Experimental results yield convincing evidence of the effectiveness and data-efficiency of our proposed framework.

In summary, we make the following contributions:

- We explore the task of manipulating articulated objects within environment constraints and formulate the task of **environment-aware** affordance learning for manipulating 3D articulated objects, incorporating **object-centric** per-point priors and environment constraints.
- To tackle the **combinatorial explosion** problem in scene complexity, we propose a data-efficient framework capable of training on scenes featuring a single occluder and generalizing to scenes with complex occluder combinations, leveraging contrastive learning and point-level local significant part representations.
- We establish benchmarking multi-object full-robot (as opposed to flying grippers) environments in SAPIEN simulator. Results show our framework learns environment-aware affordance generalizable well to novel scenes with complex novel occluder combinations.

## 2 Related Work

### 2.1 Visual Affordance for Robotic Manipulation

Visual affordance [7] is a kind of representation that indicates possible ways for robots to interact with the target and complete tasks. Many works study affordance for the classic grasping task in robotics [24, 28, 3, 17, 16, 50], while there exist many current works on point-level affordance indicating object geometrics for articulated object manipulation [25, 44, 42, 5, 6, 29], dual-gripper collaboration [54] and object to object interaction [26]. Tracing back to Gibson [7], the proposer of affordance, however, affordance notates the opportunities of interactions that involve consideration of the environment constraints and the robot, such as the occluders and the robot morphology, and thus ought to be aware of the robot and environment. Therefore, we propose the environment-aware affordance learning task, incorporating both the object actionable priors and environment constraints.

### 2.2 Occlusion Handling

Occlusion is a significant challenge for current computer vision and robotics systems. In computer vision, there have been previous works trying to handle occlusion in different tasks, including object detection [39, 14, 52], instance segmentation [48, 14, 53] and tracking [9]. Besides, to intrinsically handle the occlusion, another series of work aim to recover the entire shape of occluded objects, *i.e.*, amodal segmentation [55, 11, 52].

In robotics, the occlusion problem mainly exists in object retrieval [20, 12], grasping [40, 51, 36] or rearrangement [37, 2, 21] in clutters. In our paper, we study the task of learning the manipulation of 3D articulated objects with cluttered occlusions in front of the target object, which involves consideration of not only the observations of occlusions and the target, but also the agent's relationship with them.

# 3 Problem Formulation

We formulate a new challenging task, **Environment-Aware Affordance**, aiming to infer point-level affordance to indicate the actionable information for a full robot arm in different positions to manipulate 3D articulated objects under diverse occlusions. While previous point-level affordance learning works only consider flying grippers [25, 42, 44, 54] or fixed arms [6] to manipulate the target object without occluders, we further consider the robot location as well as scene occlusions.

Specifically, in our formulated task, given as input a 3D scene point cloud $S$ with $n$ points $p_1, p_2, ..., p_n$, containing a target articulated object $T$ with $m$ points $T_{p_1}, T_{p_2}, ..., T_{p_m}$, $k$ occluder object point cloud $O_1, O_2, ..., O_k$, and the robot position $R \in \mathbb{R}^3$, the model is required to predict the point-level affordance score $a_{T_{p_i}|R,S}$ on each target point $T_{p_i}$ of $T$, indicating how likely the point is interactable. The point-level affordance is able to guide 3D articulated object manipulation tasks.

As the complexity of occluder combinations grows exponentially, making it difficult and time-consuming to collect enough occluder combinations for training, we propose a data-efficient method that trains on only a few scenes with a single occluder, which can generalize to scenes with multiple occluder combinations.

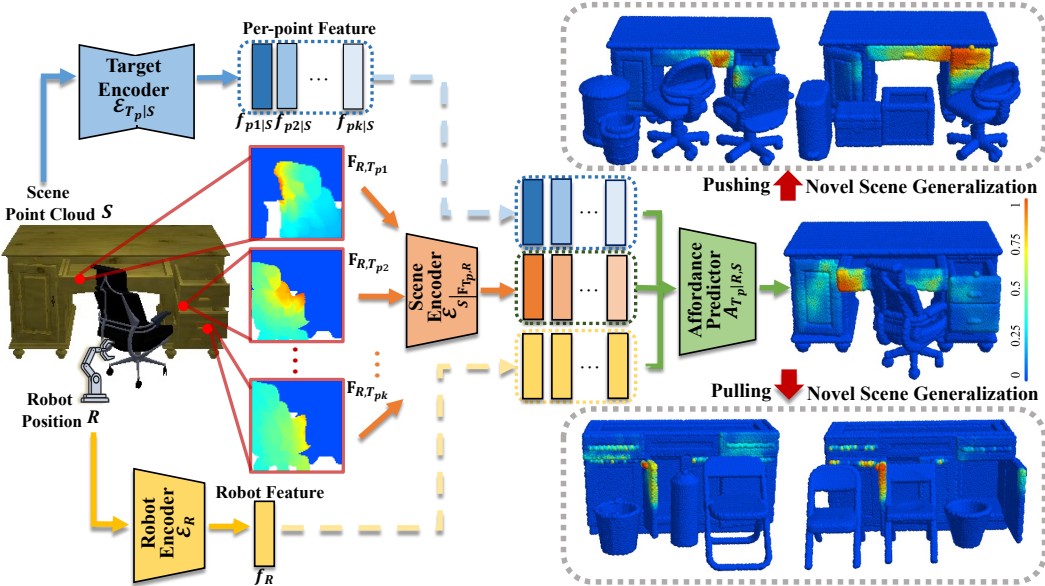

Figure 2: **Our Proposed Data-efficient Framework for Learning Environment-Aware Affordance.** Our model takes an occluded scene point cloud and robot position as input. Our framework generates per-point occlusion fields indicating most significant local parts of occluders. Then, it predicts per-point affordance using extracted features of each target point, its corresponding occlusion field, and robot position. The trained model can generalize to novel multi-occluder scenes. Red points denote manipulation points over the target object.

# 4 Method

## 4.1 Overview

Our framework is composed of four networks: Robot Encoder $\mathcal{E}_R$, Target Encoder $\mathcal{E}_{T_p|S}$, Scene Encoder $\mathcal{E}_{S|T_p,R}$ and Affordance Predictor $A_{T_p|R,S}$. While $\mathcal{E}_R$ and $\mathcal{E}_{T_p}$ extracts the robot and target representations $f_R$ and $f_{T_p|S}$ (Sec. 4.2), the most important component of our framework is learning the robot-target conditioned scene representations that are sensitive to the significant local parts in the scene for manipulation (Sec. 4.3), which uses $\mathcal{E}_{S|T_p,R}$ to extract the scene representations $f_{S|T_p,R}$ from a designed robot-target conditioned occlusion field. This component makes the model sensitive to robot-target point conditioned significant parts and thus empowers the model with the generalization in novel complicated scenes. Finally, $A_{T_p|R,S}$ takes $f_R$, $f_{T_p|S}$ and $f_{S|T_p,R}$, and predicts the point-level environment-aware affordance score $a_{T_p|R,S}$ on $T_p$ (Sec. 4.4).

## 4.2 Robot and Target Representations

Both the robot position and the target manipulation point (including its position and local geometry) will affect the environment-aware affordance. For example, when the robot position changes, the affordance will change according to the robot's new reach area and interaction modes with occluders. Similarly, when the target manipulation point changes, the originally interactable point will become non-interactive for various reasons, such as robot being unable to reach the new point, the local geometry of the target point not supporting the desired action (*e.g.*, a smooth surface does not support pulling), or the robot colliding with occluders while attempting to move to the new point. So we design a Robot Encoder $\mathcal{E}_R$ and a Target Encoder $\mathcal{E}_{T_p|S}$ to extract relevant information.

As shown in Figure 2, we employ a MLP network as $\mathcal{E}_R$, which encodes the robot position $R$ into a latent representation $f_R \in \mathbb{R}^{128}$ and a Segmentation-version of PointNet++ [31] that encodes the scene point cloud into per-point features, where $f_{T_p|S} \in \mathbb{R}^{128}$ represents the feature of $T_p$.

## 4.3 Robot-Target Conditioned Scene Representations

In scenes with multiple occluders, the quantity, geometry and arrangement of occluders demonstrate rich diversity. However, given the robot position and the target manipulation point, most points in the scene are unlikely to significantly affect the manipulation, and the area of the significant parts that do have an impact (*e.g.*, the area around occluders where potential collisions may occur during robot movement) is usually not large. By focusing on these significant parts of the scene, the change of occluder number and geometry will not matter, and a model can effectively learn to generalize to more complex scenes with multiple novel occlusion combinations.

To this end, we design the robot-target conditioned occlusion field to facilitate learning significant scene representations conditioned on the robot and the target point (Sec. 4.3.1), and propose the robot-target conditioned contrastive learning to further enhance learned scene representations (Sec. 4.3.2).

### 4.3.1 Robot-Target Conditioned Occlusion Field

Although a scene may contain a large number of points (of occlusions and the target object), when we know the target manipulation point and the robot position, the point number of the significant part is not large, and the distance between these points and either the robot or the target manipulation point is not far. Therefore, we design the robot-target conditioned occlusion field that maps the scene into a vector field conditioned on both the target point and the robot, facilitating learning scene representations highlighting significant parts for representations.

We define the *Occlusion Field* as a conditional continuous vector field $\mathbf{F}$ on an open and connected set $\mathbf{G} = \mathbb{R}^3 \setminus \mathbf{Object} \subset \mathbb{R}^3$, which can be represented by a value function $\mathbf{F}_{R,T_p} : \mathbf{G} \to \mathbb{R}^3$,

$$
\begin{aligned}
\mathbf{F}_{R,T_p}(x,y,z) &= \mathbf{V_R} \times \mathbf{V_{T_p}} \\
&= \langle x - x_R, y - y_R, z - z_R \rangle \otimes \langle x - x_{T_p}, y - y_{T_p}, z - z_{T_p} \rangle \\
&= \langle F_1, F_2, F_3 \rangle . \forall (x,y,z) \in \mathbf{G}.
\end{aligned}
\tag{1}
$$

The field factors $\mathbf{V_R}$ and $\mathbf{V_{T_p}}$ are seperately conditional on $R$ and $T_p$. For any point $(x,y,z)$ in $\mathbf{G}$, its value vector $\mathbf{F}_{R,T_p}(x,y,z)$ is calculated as the cross product ($\otimes$) of the Euclidean vectors from $(x,y,z)$ to robot $R$ and target $T_p$, two fixed points of $\mathbf{F}_{R,T_p}$. Since the field is continuous, we have:

$$
\lim_{p \to T_p} \mathbf{F}_{R,T_p}(x,y,z) = \lim_{p \to R} \mathbf{F}_{R,T_p}(x,y,z) = 0.
\tag{2}
$$

So the model can filter the unimportant points whose field values are too large.

We employ a PointNet [30] network as the Scene Encoder $\mathcal{E}_{S|T_p,R}$. It takes as input the conditional vector field $\mathbf{F}_{R,T_p}$ mapped from the scene point cloud $S$ selected with small field values, and then outputs the robot-target conditioned scene representations $f_{S|T_p,R}$.

### 4.3.2 Robot-Target Conditioned Contrastive Learning

The above-learned robot-target conditioned scene representations $f_{S|T_p,R}$ should be sensitive to the meaningful local parts of the occluders for manipulation, while agnostic to unimportant occluders and their local parts, despite the number, geometry and combinations of them. Specifically, the learned

scene representations should have the following properties: (1) given the same target point, when a new occluder occurs in the scene while not affecting the manipulation, the learned representations will keep the same; (2) given the same scene, when the target point changes, the meaningful local parts of scene as well as the learned representations will correspondingly change.

To better empower scene representations with such properties, which can further boost the performance and data-efficiency in affordance prediction, we propose the robot-target conditioned contrastive learning method for learning scene representations.

As shown in Figure 3, for predicting the point affordance $\hat{a}_{T_p|R,S}$ for $T_{R,S}(p) \in \mathcal{X}_{\mathcal{R},\mathcal{S}}$ task in single-occluder scene $S$, the target point $T_p$ and the robot $R$, we correspondingly generate a positive and a negative task. Specifically, we generate the task of predicting the point affordance $\hat{a}_{T_{p'}|R,S}$ with the same scene $S$ drawn from augmentation distribution $A(\cdot \mid T_{R,S}(p))$, robot $R$ and a different target point $T_{p'}$ drawn from marginal distribution $A(\cdot) = \mathbb{E}_{T_{R,S}(p)} A(\cdot \mid T_{R,S}(p))$, for the reason that although the scene keeps unchanged, the target point changes and thus the significant part in occluders will correspondingly change, and thus the scene representations should change. Also, we generate the task of predicting the point affordance $\hat{a}_{T_p|R,S'}$ with the same robot $R$, target point $T_p$, and a new scene $S'$ which adds to the original scene $S$ with an occluder that is uninfluential to the manipulation. Even though the scene changes from $S$ to $S'$, the significant part in the occluders keeps unchanged, and thus the target-robot conditioned scene representations should keep the same. We leverage this prior to conduct our contrastive learning, which performs better representation learning for the occlusion field.

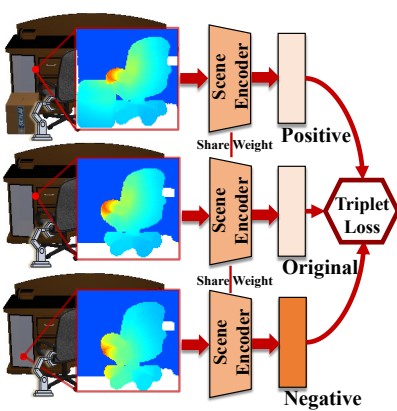

Figure 3: **Robot-Target Conditioned Contrastive Learning.** For each point affordance prediction task (Middle, $\hat{a}_{T_p|R,S}$), we respectively generate its corresponding positive (Up, same target point with an extra occluder $\hat{a}_{T_p|R,S'}$) and negative (Down, different target point $\hat{a}_{T_{p'}|R,S}$) paired tasks, and use triplet loss to learn the scene representations.

The original task and its positive and negative paired tasks constitute triplets, so we use triplet loss [32] to learn the scene representations of them contrastively ($\alpha$ is the boundary constant):

$$\mathcal{L}^{CL}_{T_p|R,S} = \left\| f_{S|T_p,R} - f_{S'|T_p,R} \right\|^2_2 - \left\| f_{S|T_p,R} - f_{S|T_{p'},R} \right\|^2_2 + \alpha \qquad (3)$$

## 4.4 Affordance Prediction

We propose the Affordance Predictor $A_{T_p|R,S}$ that aggregates the information of the target, the robot and the scene to predict the point-level environment-aware affordance.

We employ a MLP as the Affordance Predictor $A_{T_p|R,S}$ that takes $f_R$, $f_{T_p|S}$ and $f_{S|T_p,R}$ as input, and predicts the affordance score $\hat{a}_{T_p|R,S} \in R$ on each target point $T_p$ conditioned on $R$ and $S$.

We apply $\mathcal{L}_1$ loss to measure the error between the affordance prediction $\hat{a}_{T_p|R,S}$ and the ground-truth affordance score $a_{T_p|R,S}$ on a certain target point $T_p$:

$$\mathcal{L}^{AFF}_{T_p|R,S} = \mathcal{L}_1(a_{T_p|R,S}, \hat{a}_{T_p|R,S}). \qquad (4)$$

The total loss for the whole framework is then defined as:

$$\mathcal{L}^{total}_{T_p|R,S} = \mathcal{L}^{AFF}_{T_p|R,S} + \lambda_{CL} \cdot \mathcal{L}^{CL}_{T_p|R,S}, \qquad (5)$$

where $\lambda_{CL}$ is a balancing coefficient.

# 5  Experiments

## 5.1  Settings

For **tasks**, we follow Where2Act [25] and use the point-level affordance predictions of pushing and pulling articulated parts (doors and drawers) as our tasks. In a scene with occluders, a target point and a robot, the ground-truth affordance score of a target point is set to be 0 / 1 when the robot fails / succeeds in pushing or pulling the target point without colliding occluders.

For **simulation and dataset**, we use SAPIEN [45] as our simulation environment, equipped with large-scale Partnet-Mobility dataset [27] and ShapeNet [1] dataset, with occluder data statistics as shown in Table 4. Besides, we use cabinets and tables as target objects, for the reason that in the real world there often exist many occluders in front of them.

For **training**, we collect interactions in one-occluder scenes. Specifically, we collect 900 successful and 900 failure interactions for pushing, and 300 successful and 2500 failure interactions for pulling. Pulling needs more failure interactions as most points are not pullable. For each data,

| Train-Cats. | All | Basket | Bottle | Bowl | Box |
|---|---|---|---|---|---|
| Train-Data | 367 | 77 | 16 | 128 | 17 |
| Test-Data | 128 | 31 | 4 | 44 | 5 |
| | | Bucket | Chair | Pot | TrashCan |
| | | 27 | 61 | 16 | 25 |
| | | 9 | 20 | 5 | 10 |
| Novel-Cats. | All | Dispenser | Jar | Kettle | FoldChair |
| Test-Data | 589 | 9 | 528 | 26 | 26 |

Table 1: **Occluder Dataset Statistics.** We use 1,084 different shapes in ShapeNet [1] and PartNet-Mobility [27], covering 12 commonly seen indoor occluder categories. We use 8 training categories (split into 367 training shapes and 128 test shapes), and 4 novel categories with 589 shapes.

we additionally collect a positive and a negative paired interaction for contrastive learning. For **testing**, we use multi-occluder scenes respectively in training category test shapes and novel categories.

For **evaluation**, to evaluate the accuracy of affordance prediction, we follow Where2Act and use **F-Score** and **Average Precision**. To evaluate predicted affordance's capability in providing actionable priors for manipulation in scenes, we introduce **Sample Manipulation Accuracy** metric $sma$:

$$sma = \frac{\text{\# successful interactions}}{\text{\# total interactions}} \quad (6)$$

The total interactions are proposed based on affordance predictions over the whole scene. We randomly adopt proposals with a confidence threshold and compute the manipulation accuracy.

## 5.2  Baselines and Ablations

We compare our method with the following baselines that learn affordance for manipulation:

- **Where2Act (W2A)** [25] that abstracts the robot arm into a flying gripper and learns point-level actionable affordance for manipulating articulated objects.
- **W2A-R** [25] that uses a robot arm instead of a flying gripper in Where2Act's setting, and adds a robot encoder in Where2Act's networks.
- **O2O-Afford (O2O)** [26] that learns point-level affordance for object-to-object interactions. We use the robot and the scene as two input objects.
- **O2O-M** that trains O2O-Afford using scenes containing multiple occluders.

To ensure fair comparison, we train the baselines using both originally generated data and their corresponding positive and negative paired data collected for contrastive learning.

Moreover, we compare our method with Collision Avoidance RTT Planner (**RTT-CA**) [19, 35, 8], a commonly used sampling-based planner for robotic manipulation, to demonstrate the capability of point-level affordance in guiding manipulation in complicated scenes.

In addition, we compare our method with two versions that respectively ablate one core component:

- **Ours w/o OF** that learns environment-aware affordance without the occlusion field.
- **Ours w/o CL** that learns environment-aware affordance without contrastive learning.

## 5.3 Results and Analysis

Table 2, 3 and Figure 4 demonstrate the superiority of our framework quantitatively and qualitatively.

Results of **W2A** (highlighting every actionable point) demonstrate the necessity in considering the robot instead of a flying gripper in more realistic settings.

Results of **W2A-R** and **O2O** (they have similar qualitative results and we show visualization of **W2A-R**) that consider robot arms demonstrate that, our framework can better recognize the novel occluder part (*e.g.* the back of foldingchair that hinders pulling the handles in the middle doors, shown in Line 2, Column 3 and 5 in Figure 4) in novel complicated scenes.

Results of **O2O-M** demonstrate that, when the amount of training data is not large, training on complex multi-occluder scenes cannot achieve performance comparable to our data-efficient method.

Results of **Ours w/o OF** and **Ours w/o CL** demonstrate the necessities of two components. With **occlusion field**, our model better recognizes target areas that may exist collision with novel-shaped occluders (*e.g.*, the drawer board hindered by the basket in Line 1, the door edge hindered by in Line 2, Figure 5). With **contrastive learning**, our model better discriminates between actionable and non-actionable points, while **Ours w/o CL** predicts reasonable affordance but with less certainty.

Comparisons between **RTT-CA** and affordance methods show that, while sampling-based planners may face difficulty in complicated scenes, point-level affordance may be more suitable by directly providing per-point actionable priors for manipulating articulated objects within complex constraints.

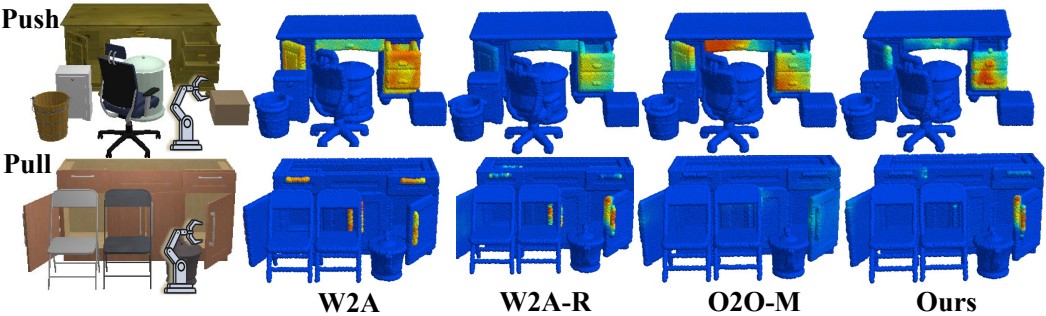

Figure 4: **Qualitative Comparisons between Our Method and Baselines.**

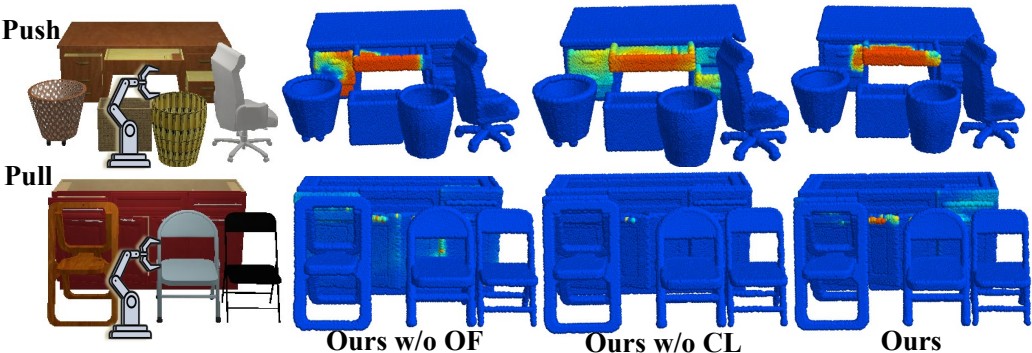

Figure 5: **Qualitative Comparisons between Our Method and Ablations.**

| Task | RRT-CA | W2A | O2O | O2O-M | W2A-R | Ours w/o OF | Ours w/o CL | Ours |
|------|--------|-----|-----|-------|-------|-------------|-------------|------|
| pushing | 23.65 | 17.80 | 34.60 | 37.33 | 34.76 | 39.43 | 36.63 | **43.52** |
| pulling | 10.31 | 3.82 | 29.93 | 29.36 | 27.55 | 33.19 | 28.02 | **36.19** |

Table 2: **Sample Manipulation Accuracy(%).**

| Method | pushing | pushing (novel) | pulling | pulling (novel) |
|---|---|---|---|---|
| W2A | 64.70 / 52.97 | 62.80 / 47.07 | 66.42 / 37.59 | 60.37 / 40.73 |
| W2A-R | 68.66 / 71.31 | 64.86 / 69.55 | 34.06 / 72.56 | 26.79 / 73.79 |
| O2O | 65.18 / 77.07 | 59.04 / 72.18 | 46.03 / 74.06 | 43.42 / 73.56 |
| O2O-M | 61.06 / 67.82 | 67.43 / 68.54 | 39.76 / 66.42 | 33.29 / 61.47 |
| Ours w/o OF | 64.28 / 69.02 | 58.31 / 66.94 | 36.36 / 67.19 | 33.33 / 64.30 |
| Ours w/o CL | 68.59 / 74.02 | 64.70 / 71.62 | 65.74 / 75.40 | 55.25 / 71.92 |
| Ours | **77.30 / 86.11** | **76.11 / 83.58** | **71.28 / 75.96** | **61.32 / 75.18** |

Table 3: **Quantitative Evaluations and Comparisons with Baselines and Ablated Versions.** In each entry, we report **F-Score(%)** and **Average Precision(%)** before and after slash.

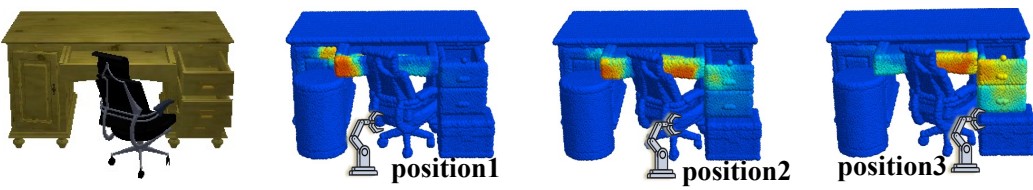

Figure 6: **Predicted Affordance Changes Conditioned on Different Robot Rositions.**

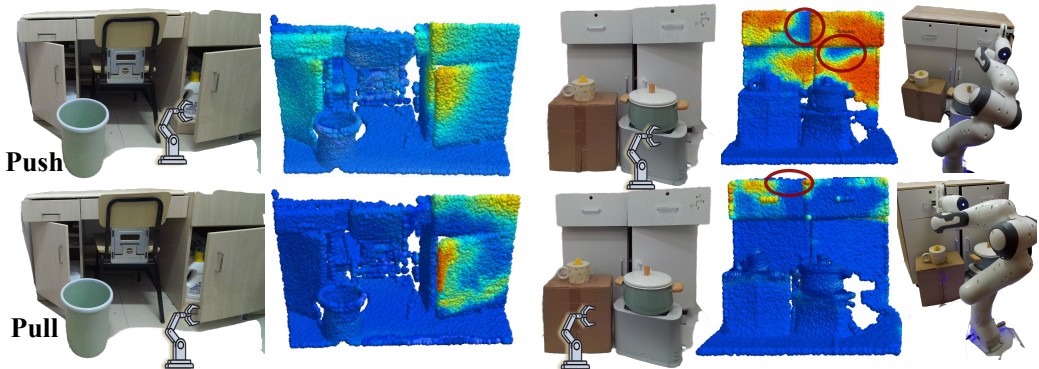

Figure 7: **Predicted Affordance on Real-World Scans.**

Additionally, Figure 6 shows our method generates different environment-aware affordance conditioned on different robot positions in the same scene. Figure 7 demonstrates our framework generates promising results by testing on different real-world scenes. It is worth mentioning, as denoted in the red circles, our model not only learns constraints from occluders, but also learns to avoid manipulating points that may lead to collisions with other parts of the object (*i.e.*, self occlusion).

# 6   Conclusion

We introduce environment-aware affordance for manipulating 3D articulated objects within environment constraints, leveraging point-level representations to address the combinatorial explosion challenge in scene complexity. Using an interactive environment built upon SAPIEN and the PartNet-Mobility and ShapeNet datasets, we train neural networks that predict per-point actionable information for manipulating articulated 3D objects under occlusions. We present extensive quantitative evaluations and qualitative analyses of the proposed method. Results show that the learned priors are highly localized and thus generalizable to novel scenes with unseen occluder combinations.

**Ethics Statement.**   Our work has the potential to enable robots on articulated object manipulation in complicated scenes. The learned environment-aware affordance avoids collisions in manipulation, and thus reduces the risk of accident. We do not see our work has any particular harm or issue.

## 7    Acknowledgment

This work was supported by National Natural Science Foundation of China - General Program (62376006) and The National Youth Talent Support Program (8200800081).

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

# Appendix

## A    More Details on Simulation and Settings

Following Where2Act [25], we design our interactive simulation environment based on SAPIEN, using the same set of simulation parameters for all interaction trials.

For **general simulation settings**, we use frame rate 500 fps, tolerance length 0.001, tolerance speed 0.005, solver iterations 20 (for constraint solvers related to joints and contacts), with Persistent Contact Manifold (PCM) disabled (for better simulation stability), with disabled sleeping mode (*i.e.* no locking for presumably still rigid bodies in simulation), and all the other settings as default in SAPIEN release.

For **physical simulation**, we use the standard gravity 9.81, static friction coefficient 4.0, dynamic friction coefficient 4.0, and restitution coefficient 0.01. For the object articulation dynamics simulation, we use stiffness 0 and damping 10.

For the **rendering**, we use OpenGL-based rasterization rendering for the fast speed of simulation. We set three point lights around the object (one at the front, one from back-left and one from back-right) for lighting the scene, with mild ambient lighting as well. The camera is set to have near plane 0.1, far plane 100, resolution 448, and field of view $35°$.

For **3D partial point cloud scan inputs**, we back-project the depth image into a foreground point cloud, by rejecting the far-away background depth pixels, and then perform furthest point sampling to get a 10K-size point cloud scan.

For **robot arm movement**, we use RRT Planner [35, 8, 19] equipped with PID controller to generate and execute a certain path towards the target.

For an **interaction trial** to be considered successful, it not only needs to cause considerable part motion along intended direction. To avoid the extreme data unbalance in pulling data, we manually set handle mask on our simulator and assign half of the interactions on the handles. To simplify the consideration of different interaction directions' impact on affordance, we set every interaction to move along the normal direction of the target point.

## B    More Data Details and Visualization

In Table 4, we summarize our data statistics. In Fig. 8, we visualize our simulation assets from ShapeNet [1] and PartNet [27] that we use in this work.

| Train-Cats | All | Basket | Bottle | Bowl | Box | Bucket | Chair | Pot | TrashCan |
|---|---|---|---|---|---|---|---|---|---|
| Train-Data | 367 | 77 | 16 | 128 | 17 | 27 | 61 | 16 | 25 |
| Test-Data | 128 | 31 | 4 | 44 | 5 | 9 | 20 | 5 | 10 |
| Test-Cats | All | Dispenser | Jar | Kettle | FoldingChair | | | | |
| Test-Data | 589 | 9 | 528 | 26 | 26 | | | | |

Table 4: **Occluder Dataset Statistics.** We use 1,084 different shapes in ShapeNet [1] and PartNet-Mobility [27], covering 12 commonly seen indoor occluder categories. We use 8 training categories (split into 367 training shapes and 128 test shapes), and 4 test categories with 589 shapes networks have never seen in training.

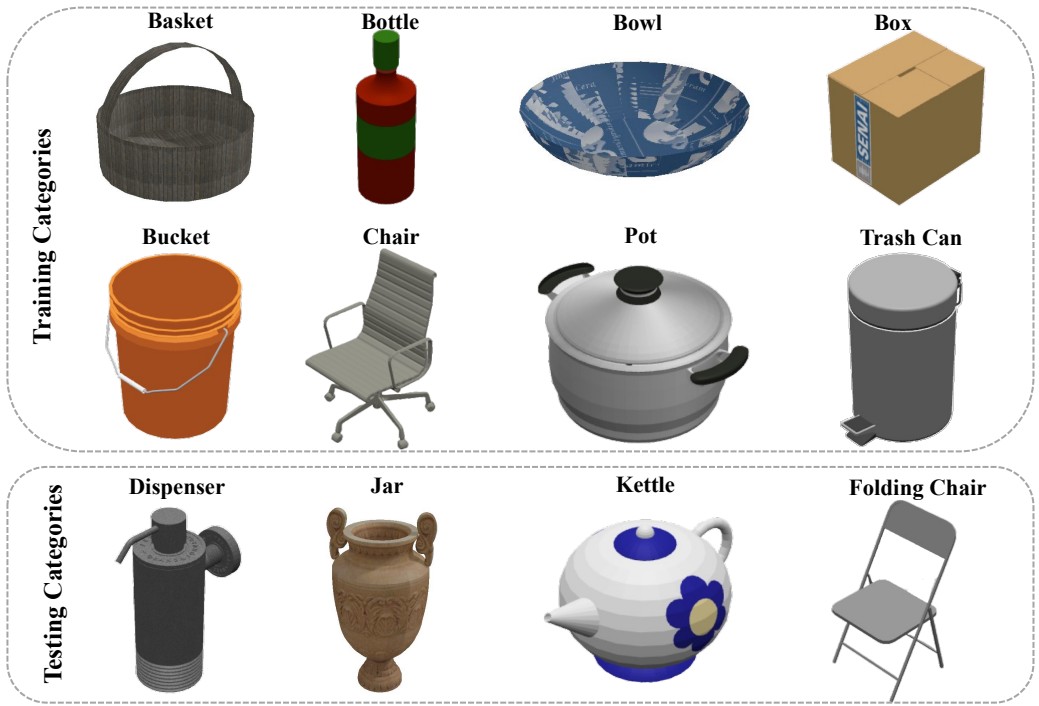

Figure 8: **Our simulation assets from ShapeNet [1] and PartNet [27].**

## C  More Training Details

### C.1  Hyper-parameters

We set the batch size to 30, and use Adam Optimizer [15] with 0.001 as the initial learning rate.

We use const 2.00 as the boundary constant in $\alpha$ contrastive learning, and 1.00 as the balancing coefficient $\lambda_{CL}$ in the total loss.

### C.2  Computing Resources

We use PyTorch as our Deep Learning framework, and single RTX GeForce 3090 (20GB GPU) for training and inference.

## D  More Results and Analysis

Fig. 9 10 11 12 demonstrate comparions with baselines and ablations. Fig. 13 shows the whole occlusion fields. Fig. 14 shows real-world demonstrations with analysis in the caption.

## E  Discussion and Visualization on Failure Cases

We present some interesting failure cases in Fig. 15. From these examples, we see the difficulty of the task. Also, given the current problem formulation, there are some intrinsically ambiguous cases that are generally hard for robots to figure out from a single static snapshot. Moreover, our affordance is trained on data generated with robots deploying a naive policy, *e.g.*, pushing the normal direction of the target plane. Better policies may introduce better affordance.

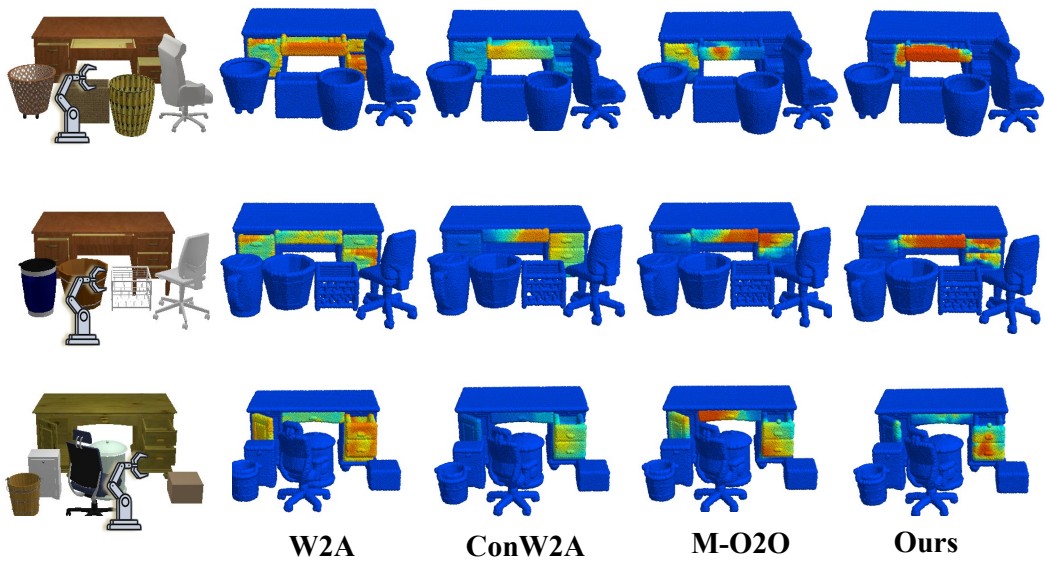

Figure 9: **More Qualitative Comparisons between Our Method and Baselines in Pushing.**

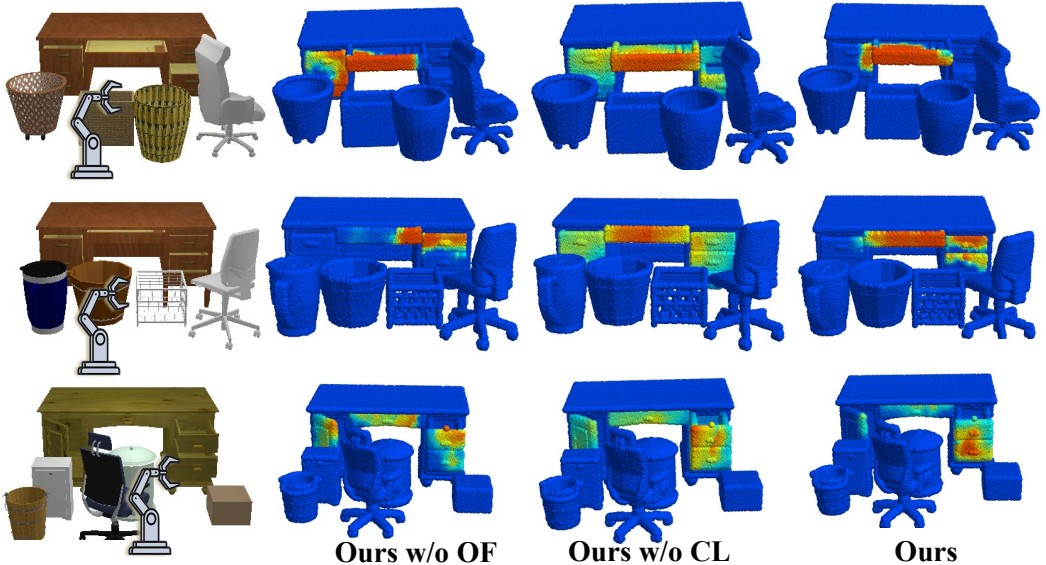

Figure 10: **More Qualitative Comparisons between Our Method and Ablations in Pushing.**

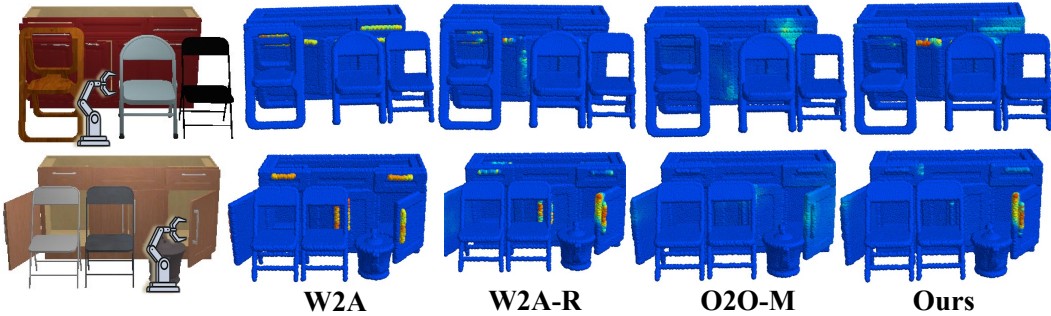

Figure 11: **More Qualitative Comparisons between Our Method and Baselines in Pulling.**

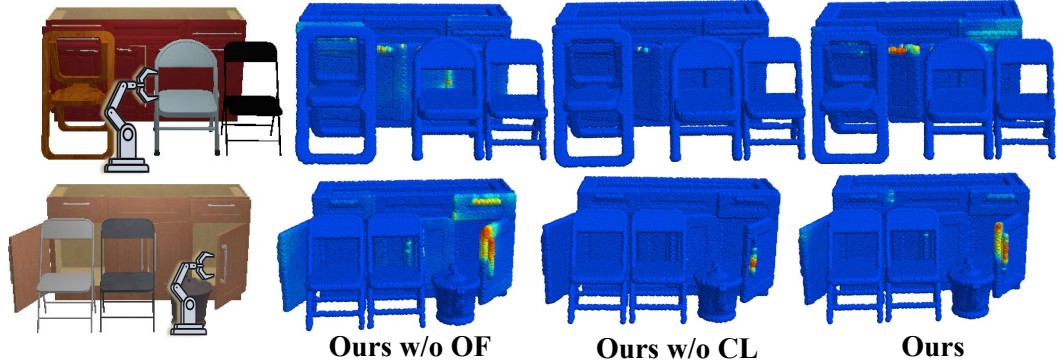

**Ours w/o OF**  **Ours w/o CL**  **Ours**

Figure 12: **More Qualitative Comparisons between Our Method and Ablations in Pulling.**

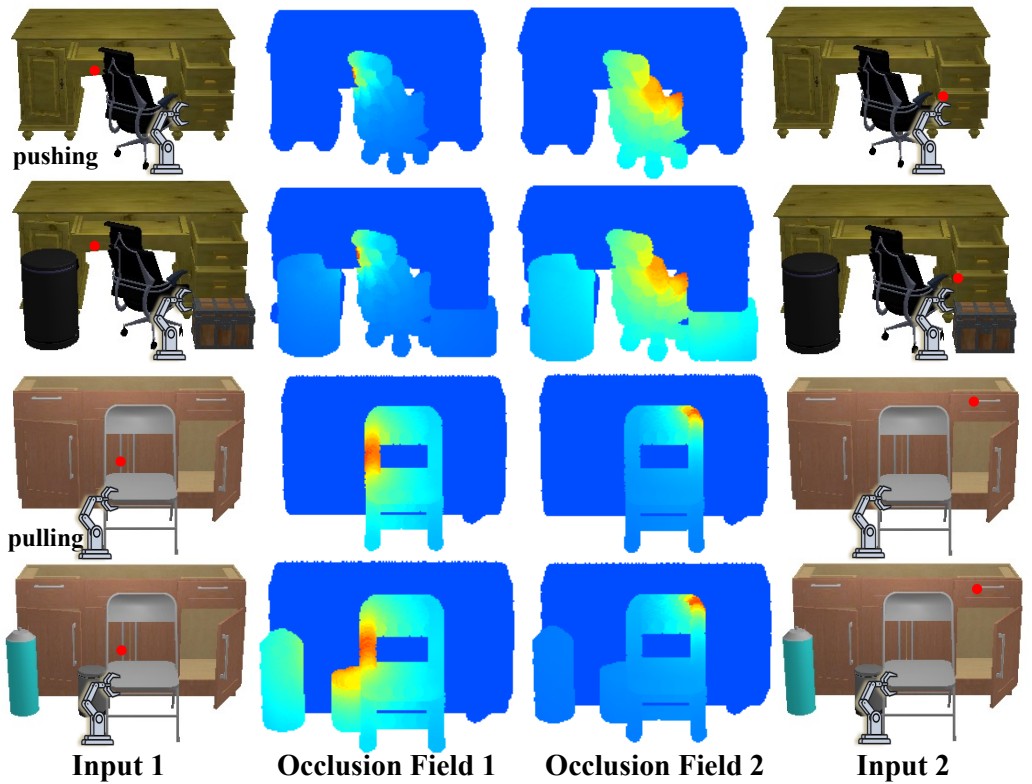

**Input 1**  **Occlusion Field 1**  **Occlusion Field 2**  **Input 2**

Figure 13: **Visualization of the whole Occlusion Fields.**

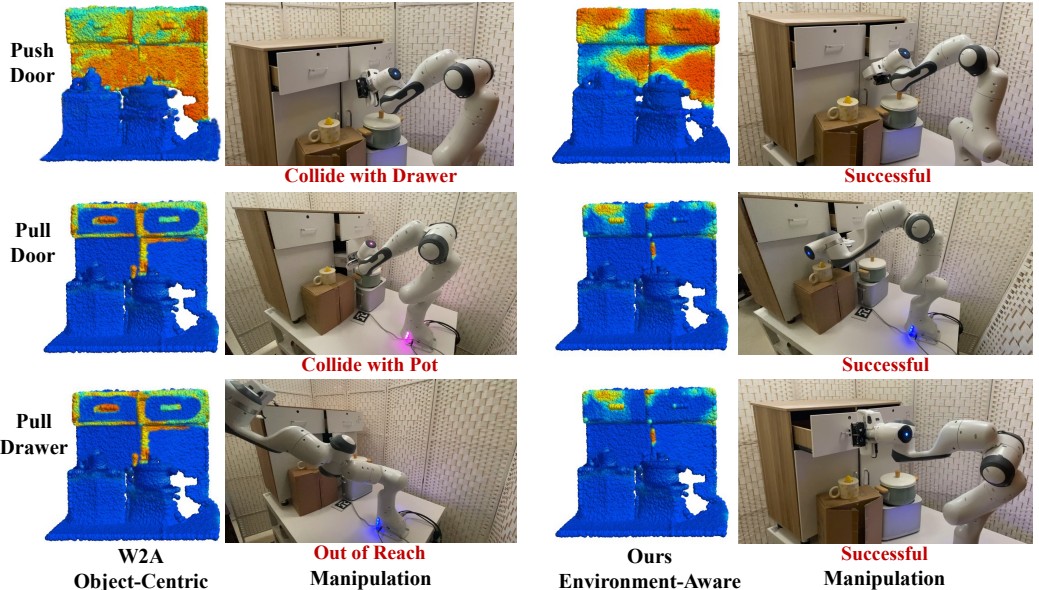

Figure 14: **Real-World Demonstrations of Manipulation Policy Guided by Object-Centric Affordance and Our Proposed Environment-Aware Affordance.** It is clear that environment-aware affordance can help avoid out-of-reach situations and collisions with other self-parts or objects.

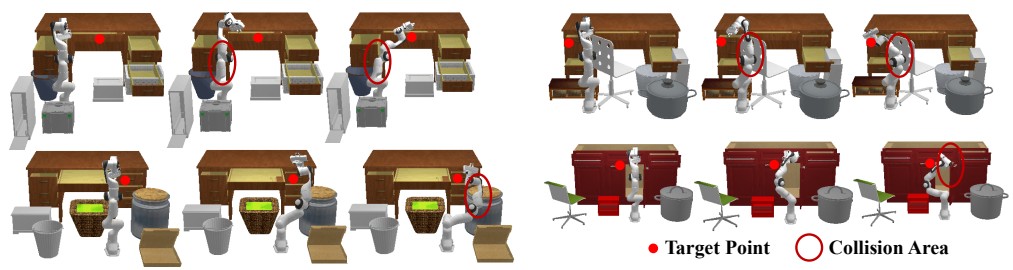

Figure 15: **Failure cases**. Up: robot initialized close to occluders. Down: Motion planning method leads the robot to collide with occluders when approaching a plausible target point.

## F  Future Work on Robot-Target Conditioned Contrastive Learning

Limited to simulator configuration, our contrastive learning method only considers a limited augmentation distribution $A(\cdot \mid \bar{x})$ for each anchor scene $\bar{x} \in \mathcal{X}$ while the marginal distribution $A(\cdot) = \mathbb{E}_{\bar{x}} A(\cdot \mid \bar{x})$ is complete. The augmentation distribution $A(\cdot \mid \bar{x})$ only includes one more occluder at the edge of $\bar{x}$, and neglects the potential augmentation methods by choosing similar target points. Future methods can be applied with a better similarity metric of comparing different things and improve our self-supervised learning paradigm.

