# OpenReview forum: "Learning Environment-Aware Affordance for 3D Articulated Object Manipulation under Occlusions"
_NeurIPS.cc/2023/Conference — NeurIPS 2023 poster_

### Official Review · Reviewer_wNB2 · 2023-06-29

**Soundness:** 3 good
**Presentation:** 2 fair
**Contribution:** 3 good
**Rating:** 6
**Confidence:** 4

**Summary:**

This work formulates environment-aware affordance for articulated object manipulation, which incorporates both object-level per-point actionable priors and environment constraints. First, this work identifies limitations of previous works on object-level affordance learning, including over-simplified robot manipulators and environment occlusion; it proposes to incorporate robot and environment constraints into the per-point affordance score. Next, this work proposes a contrastive learning method to allow the learned affordance prediction model to generalize to complex and unseen occluder combinations. This work evaluates the success rate of the proposed method with open door and drawer tasks in simulated environments and show improvements over previous affordance learning methods, and it demonstrates that the learned affordance can be useful in real robotic tasks by showing real robot demos.

**Strengths:**

- This work has a clear goal and motivation. It addresses several important limitations of previous affordance learning works (over-simplified robots and environments), allowing per-point affordance prediction to be deployed on real robots to complete tasks in more complex and realistic environments.

- This work provides adequate quantitative and qualitative experiments and comparison with baselines to show the effectiveness of the proposed method. It also justifies the necessity of 2 significant components, occlusion field and contrastive learning, with ablation studies.

**Weaknesses:**

1. This paper does not provide a clear definition of per-point affordance in problem formulation. I think this paper needs to provide a more comprehensive definition of "affordance" and "interactable" in introduction or section 3, preferably with concrete examples. Currently the motivation and goal of this paper is hard to understand without reading a previous work such as Where2Act.

2. The problem formulation is also missing several key points.

    - What is the scene point cloud? Based on the experiment section, it seems to be a colored point cloud obtained from a depth camera. This information needs be explained clearly in the formulation.
   - How is the point cloud segmented into target point cloud and occluder point cloud? This information is available in the simulation, but how is it done in the real-world experiments?

3. The robot-target conditioned occlusion field is not described clearly. I find section 4.3.1 very confusing. The occlusion field is defined on a open and connected set $S$, but I cannot see how $S$ is constructed. I assume that $S$ is the space left after *filtering the unimportant points whose field values are too large*, but the field is constructed with cross-products, and it is not clear which norm is used to define *large*. Moreover, $S$ clash with the notation of the scene point cloud (which is clearly not open and not connected). I believe this section needs to be reworked to provide a clear explanation.

**Questions:**

1. How are the simulation assets pre-processed? What convex decomposition is used?

   - With default PartNet-Mobility and SAPIEN simulator, all assets would be processed into convex shapes and form a large gap between physics and rendering. Manipulation experiments performed on such data would not reflect a realistic scenario. So it is important to describe the data pre-processing method.

2. Why robot position $R$ is used as input instead of transforming the point cloud? Is there some important choice on canonical frame that is not described in the text? How do you ensure the model prediction is invariant to a rigid transformation of the entire scene?

3. What are the setup for the real-world experiments? How are the models trained and deployed? Is there finetuning on real-world data? Is there domain gaps when the model is trained on perfect point clouds in simulation and deployed on noisy data in the real world?

4. What is the inference time for the method when deployed on real scenes?

**Limitations:**

The paper provides a description of some limitations and I have raised questions of other potential limitations in the previous *Questions* section.

---

> ### Author Rebuttal · Authors · 2023-08-10
>
> Thanks you! We have provided responses below.
>
> > A clear definition of per-point affordance in problem formulation.
>
> Thanks for your suggestion! We follow [1] to define per-point affordance as an actionability score of each point on the object. Interactable means literally movable towards the intended direction to a threshold when interacted by a robot without collision. We will further provide more details in the introduction and problem formulation.
>
>
> > Clarification on the scene point cloud.
>
> Following settings of [1, 2, 3, 4], we use point clouds (without color) obtained from a depth camera as scene point cloud. Thanks for this comment and we will further clarify it in problem formulation (line 118).
>
>
> > Segmentation mask.
>
> Please see the **Clarification on the segmentation mask** Section in **Global Response TEXT** for clarification.
>
>
> > Clarification on robot-target conditioned occlusion field.
>
> Sorry for making you confused. For the **definition and property** of the robot-target conditioned occlusion field $F$, $F$ is defined in the whole $R^3$ space excluding the object $O$ (i.e., $R^3$\\ $O$). We query each occlusion point $P \in R^3$ \\ $O$ to get field values on the occlusions, and use the queried results to filter out the unimportant points in the scene.
> Thus, we notate the robot-target conditioned occlusion field as open and connected since $R^3$ \\ $O$ is open and connected, not that the space that occlusion objects possess is open and connected.
> For the **NORM to define field value**, our field value is defined as the NORM of the vector of cross-products. We use this to filter the unimportant points whose field values (NORM of the vector of cross-products) are too large.
> Thanks again for pointing out and we will refine our expression. To avoid the clash with the notation of the scene point cloud $S$, we will use symbol $G$ to notate the space $R^3$ \\ $O$  in our refined paper.
>
> > Convex decomposition to pre-process simulation assets.
>
> To reduce the gap between physics and rendering, we use the Voxelized Hierarchical Approximate Convex Decomposition (V-HACD) algorithm to decompose the 3D surface into a set of "near" convex parts, which is also used in the affordance learning work [5]. Specifically, V-HACD involves the following steps: 1. convert the 3D model into a voxel representation; 2. progressively merge voxels into larger ones while retaining the essential geometric features; 3. transform the merged convex voxels into convex polygons to further simplify the representation. After this procedure, the gap between physics and rendering is reduced and the manipulation can be conducted well. Thanks for this question! We will add more descriptions in the paper.
>
> > Robot position R v.s. transforming the point cloud. More details about the input point cloud.
>
> We take **robot position R** as input instead of transforming the point cloud because we want that the model can be deployed before a robot is observed in a scene, and thus we can pre-choose different positions for predictions and find a suitable one, which is essential for future work of movable robots and arm-based cameras. Our canonical frame is based on the target object’s center for convenience. For rigid translation, since we normalize the scene point cloud centered at the object’s center, our model prediction is invariant to rigid translation of the entire scene. For rigid rotation, since point clouds of different perspectives can be rotated into the same perspective with differences in the vacancy of point clouds given camera intrinsics and extrinsics, our method is robust to the complexity of perspectives (Figure C in the **Global Response PDF** provides affordance predictions on point clouds scanned from different perspectives with promising consistency).
>
>
> > Setup and more details of the real-world experiments.
>
> For **point cloud**, we use Microsoft’s Azure Kinect Depth Scanner to scan the depth image of the scene, and project the depth image to point cloud. Instead of fine-tuning our model on real-world data (collecting real-world data in diverse scenarios is costly), we directly deploy our model pre-trained on prefect point clouds in real-world point clouds, and get promising affordance. A series of papers have demonstrated point-level affordance learned in simulation is robust and can generate promising results when directly deployed on real-world point clouds (one possible reason: the noises do not influence a lot on the actionable information reflected by the local geometry of each point, which is what point-level affordance focuses on): Figure 6 in [1], Figure 5 in [2], Figure 6 in [3], Figure 1, 7 in [4] and Figure 8 in supplementary in [4]. Moreover, compared to the ZED-MINI Scanner and RealSense Scanner used in [2, 3, 4], point clouds scanned by our Microsoft’s Azure Kinect Depth Scanner are less noisy (which can be observed from the scanned point clouds shown in Figure 7 in our paper and scanned point clouds shown in above-mentioned figures in [2, 3, 4]), and thus the sim2real gap is smaller for our work on real-world point clouds. For **robot and control**, we use Franka Panda as the robot arm (shown in Figure 7 in the main paper, Figure 7 in supplementary pdf and 3:35 - 4:09 in the video), the same as the robot in simulation. We use the Robot Operating System (ROS) to control the robot.
>
> > Inference time when deployed on real scenes.
>
> Our inference is very quick thanks to the wonderful architecture of PointNet++. It only takes 0.18s for one scene inference on average.
>
>
> [1] Where2Act: From Pixels to Actions for Articulated 3D Objects. ICCV, 2021.
>
> [2] VAT-Mart: Learning Visual Action Trajectory Proposals for Manipulating 3D ARTiculated Objects. ICLR, 2022.
>
> [3] AdaAfford: Learning to Adapt Manipulation Affordance for 3D Articulated Objects via Few-shot Interactions. ECCV, 2022.
>
> [4] DualAfford: Learning Collaborative Visual Affordance for Dual-gripper Object Manipulation. ICLR, 2023.

---

> > ### Comment · Reviewer_wNB2 · 2023-08-14
> >
> > I would like to thank the authors to provide detailed responses for my questions. The rebuttal has addressed most of my concerns. I do not find major issues after reading the clarification on the design of segmentation mask, robot-target conditioned occlusion field, and real-world setups. Therefore, I still **recommend to accept** this paper.

---

> > > ### Author Response · Authors · 2023-08-21
> > > **Thank you!**
> > >
> > > We are glad that our responses help address your concerns. Thanks for your positive rating and recommendation to acceptance!

---

### Official Review · Reviewer_ntD4 · 2023-07-04

**Soundness:** 3 good
**Presentation:** 3 good
**Contribution:** 2 fair
**Rating:** 5
**Confidence:** 4

**Summary:**

The paper introduces the environment-aware affordance task, taking into account both object-centric affordance and environmental constraints. More specifically, the proposed task is designed to incorporate various occlusion cases and the robot's position.  In order to tackle the inherent challenge of combinatorial explosion, the paper introduces a novel framework that utilizes a local robot-target conditioned occlusion field and contrastive losses. This framework enables effective training on simpler scenarios with only one occluder, while also facilitating generalization to more complex scenes.

**Strengths:**

* The proposed task that takes the environment constraints into account is novel and insightful. Occlusion and robot position conditioned affordance is intuitive and potentially useful for downstream robot tasks.
* The idea of utilizing a local robot-target occlusion field to enable generalization to more complex occlusion scenarios is straightforward and effective based on the experimental results.
* The strategy employed to construct positive and negative pairs for the contrastive loss is intriguing, and the ablation studies confirm its efficacy in improving the scene encoder.

**Weaknesses:**

* In the proposed environment-aware affordance task, both the segmentation of the target object and the occluded objects are provided. For previous work like W2A, which only consider the object-centric affordance, it’s unclear whether the addition of simple heuristics can yield positive results. One possible heuristic is to examine the predicted affordance from W2A and determine if there are any points close to the points of occluder, subsequently filtering the affordance based on this observation. While I acknowledge the significance of environment-aware affordance for downstream tasks, I remain uncertain about the complexity and realism of the current proposed task. In addition, for the target objects, there are only cabinets and tables, it’s also unclear if the train and test scenario uses the same instance or different instances from the same category. Only two categories may not be challenging enough for the task.
* For the robot position, it’s unclear how many positions are chosen in the train and test set. From the supplement video, there seems to be 4 different robot positions. It’s unclear if the method can generalize to new robot positions well.
* For the push and pull action, it’s unclear if the default action direction is the surface normal. There should be more descriptions on the action selection and the dataset construction, including how to pick the occluder, how to add the occluder and how to judge if the action succeed considering the occluded objects.

**Questions:**

* What’s the performance of the trained model on scenes without any occlusion? And what will be the performance compared to the prior work in this scenario?
* When inference, is the segmentation of the target object and all occluded objects are given? For the real case, how do you get the segmentation?
* Is the number of points of the target object fixed for the network? How about the number of points for the occluded objects? If the number of points is not fixed, how to balance the number of points in each train example?

**Limitations:**

The authors mention the limitation.

---

> ### Author Rebuttal · Authors · 2023-08-10
>
> Thanks for valuable comments! We have addressed them below. Hope to hear back from you if you have further questions!
>
> > Whether the addition of simple heuristics can yield positive results.
>
> We have conducted the corresponding Where2Act with the heuristic experiment with the successful rates of *28.35%* in pushing, while our method achieves a successful rate of *43.52%*.
> The reason for such heuristic method’s failure is, only the distance between occlusions and the point on which the gripper interacts with the object could not fully avoid the whole robot’s collisions. Specifically, as the robot is composed of many links connected by joints, when the gripper is moving to the target point, the movements of other links are not free, and those links may have different poses and touch different occlusions in moving. In other words, in the proposed Where2Act+heuristic, even though the final pose of the gripper could avoid collision with occlusions, during the movement of the robot, chances are that some part of the robot may touch the occlusions.
>
>
> > Clarifications on instances and categories in training and test, and challenges of our setting.
>
> The main focus and contribution of our work is to learn affordance with environment constraints, thus the generalization is **mainly demonstrated in the environment constraints**, as the generalization of point-level affordance towards more target object categories has been comprehensively demonstrated in previous works[1, 3, 5]. As for our setting, while trained on 8 kinds of 367 different occluders, our model is tested on combinations of 128 **unseen objects** in those categories and 589 objects in 4 **unseen categorie**s in the scene. Please see statistics and example occluders in Table 1 in main paper and Figure 1 in the supplementary PDF.
> Moreover, the model is trained on **scenes with only one occluder**, while tested on scenes with **combinations of different numbers and kinds of objects**, which can demonstrate the data efficiency of our framework.
> Therefore, the test setting is actually quite challenging.
> As for the instances and categories of the **target object**, the chosen two categories (Cabinets and Tables) are the most complicated categories with various self-occlusion parts in the SAPIEN simulation. Inferring environment aware affordance on Cabinets and Tables needs not only consideration of the interaction constraints from occluders and robots, but also the measure of self-occlusion from different parts of the object itself that may affect the target part. Besides, in the real world there often exist many occluders in front of them.
> Additionally, we have trained a model on **three new categories** with good affordance predictions (shown in Figure B in the **Global Response PDF**), showing that the framework can handle scenes with more diverse target objects.
>
> > How many positions are chosen in the train and test set.
>
> The robot position in our setting is continuous in $R^3$ for both training and test, and our method can generalize to robot positions within reach and out of reach of the target.
> Besides, Figure 6 in the main paper shows affordance when robots are in more different positions.
>
> > Clarification on action directions, more descriptions on the action selection and dataset construction.
>
> We used the surface normal as the action direction since it can work in most cases and our work mainly focuses on occlusion and robot constraints instead of action directions.
> As for more descriptions of action selection and the dataset construction, we followed the setting of [1]. The dataset is constructed self-supervisedly in SAPIEN. Specifically, the occluder is randomly chosen from Table 1 in the main paper, and randomly added in a wide area (X: (-1.0m, 1.0m); Y: (0, 1.5m)) in front of the target object (0, 0). An action is judged to be successful when it successfully moves the target part at a threshold (greater than 0.01 unit-length or 0.5 relative to the total motion range of the articulated part) without colliding with the occluded objects.
> Due to the page limit, we did not include many such descriptions in the submission, and we will add more descriptions in the main paper and supplementary. Thank you for this comment!
>
>
>
> > What’s the performance of the trained model on scenes without any occlusion? And what will be the performance compared to the prior work in this scenario?
>
>
> We have conducted the corresponding experiment and show the successful rates of our method and previous works on the pushing task: *W2A / O2O / Ours: 40.29% / 43.37% / 63.42%*.
>
> The results show that our model performs well on scenes without any occlusion.
> Even though it is a simple setting, prior object-centric works (Where2Act, O2O-Afford) do not consider the robot arm’s constraint so they still fall behind us.
>
> > When inference, is the segmentation of the target object and all occluded objects are given? For the real case, how do you get the segmentation?
>
> See **Clarification on the segmentation mask** Section in **Global Response TEXT**.
>
> > Number of points of the target object.
>
> The number of points of the target object and that of the occluded objects are not respectively fixed. Instead, we fix the number of points of the whole scene to be 30000.
> Specifically, we use Furthest Point Sampling (FPS) to sample 30000 points from the scanned point cloud.
>
> [1] Where2Act: From Pixels to Actions for Articulated 3D Objects. ICCV, 2021.
>
> [2] O2O-Afford: Annotation-Free Large-Scale Object-Object Affordance Learning. CoRL, 2021.
>
> [3] VAT-Mart: Learning Visual Action Trajectory Proposals for Manipulating 3D ARTiculated Objects. ICLR, 2022.
>
> [4] AdaAfford: Learning to Adapt Manipulation Affordance for 3D Articulated Objects via Few-shot Interactions. ECCV, 2022.
>
> [5] DualAfford: Learning Collaborative Visual Affordance for Dual-gripper Object Manipulation. ICLR, 2023.
>
> [6] PointNet++: Deep Hierarchical Feature Learning on Point Sets in a Metric Space. NeurIPS, 2017.

---

> ### Author Response · Authors · 2023-08-18
> **Looking Forward to Seeing Your Response!**
>
> Dear reviewer ntD4,
>
>
> As the discussion phase is quickly passing, we want to know if our response resolves your concerns. If you have any further questions, we are wholeheartedly more than happy to discuss them. We greatly appreciate the invaluable suggestions you've provided!
>
>
> Best,
>
> All anonymous authors

---

> > ### Comment · Reviewer_ntD4 · 2023-08-18
> >
> > Thanks for the responses from the authors. The responses have resolved most of my questions. I'd like to raise my score

---

### Official Review · Reviewer_zyYG · 2023-07-05

**Soundness:** 2 fair
**Presentation:** 3 good
**Contribution:** 2 fair
**Rating:** 6
**Confidence:** 3

**Summary:**

Given a 3D point cloud and a robot, the method predicts per-point actionable affordance. More specifically it predicts whether a given point is pushable or pullable by the robot in a given position. The method takes into account environment occlusion. The paper presents a data-efficient training framework that allows training on a single occlusion and generalize to multiple occlusions. In addition, the paper presents a novel contrastive learning approach.

**Strengths:**

- Acounting for occlusion when predicting affordance is a step up from previous work. It has not been tackled before as far as I know.
- The paper presents several novelties: problem setup and formulation, data-efficient framework, and contrastive learning approach
- The method outperforms previous work.
- Paper is well written and easy to follow.
- The results are well presented and accompanied by a good analysis

**Weaknesses:**

- The action space is too small. Only pushing and pulling are considered.
- The test setting seems too simple. Most of the results are shown on very similar desks.
- Related work section is too short and does not clearly explain what is the gap in previous work, nor how the paper is filling that gap.
- Section 4.3.1 is lacking a supporting figure
- The robot geometry is not taken into account.

**Questions:**

None

**Limitations:**

Yes

---

> ### Author Rebuttal · Authors · 2023-08-10
>
> Thanks you! We have provided responses below.
>
> > The action space is too small. Only pushing and pulling are considered.
>
> We follow the settings of a series of affordance learning works [1, 3, 4] and use pushing and pulling as the action primitives, as they are the most common action primitives, and many actions can be the combination of them. Besides, the affordance predictions on these two actions can already demonstrate the contribution of superiority of our proposed framework in learning environment-aware affordance.
>
>
>
> > The test setting seems too simple. Most of the results are shown on very similar desks.
>
> The main focus and contribution of our work is to learn affordance with environment constraints, and thus the generalization is mainly demonstrated in the environment constraints (occluder numbers, categories, shapes and combinations).
> While trained on scenes containing **only one occluder** in 8 kinds of 367 different objects, our model is tested on scenes containing **combinations of occluders** from **128 unseen objects** in those categories and 589 objects in **4 unseen categories**. Detailed statistics and example occluders are shown in Table 1 in the main paper and Figure 1 in the supplementary PDF. Therefore, the test setting is actually quite difficult, which requires the strong generalization ability towards novel complex scenes with different occluder numbers, categories, shapes and combinations.
> As for the category of the target object,  previous point-level affordance works have demonstrated the generalization ability towards different *target* objects [1, 3, 5].
> We choose two categories (Cabinets and Tables) as they are the most complicated categories with various self-occlusion parts in the SAPIEN simulation. Inferring environment-aware affordance on Cabinets and Tables needs not only consideration of the interaction constraints from occluders and robots, but also the measure of self-occlusion from different parts of the object itself (e.g., pulling a drawer without colliding with another drawer nearby). Besides, in the real world there often exist many occluders in front of them, making our setting very practical. Additionally, we have trained a model on three new categories with good affordance predictions (shown in Figure B in the **Global Response PDF**), showing that the framework can handle scenes with more diverse target objects.
>
>
>
>
> > Related work section is too short and does not clearly explain what is the gap in previous work, nor how the paper is filling that gap.
>
> Thank you for pointing out this. We will add more discussions on the gap between our work and previous works (integrating object-centric actionable priors in previous works with environment constraints in our work) and how we fill this gap (proposing robot representations, robot-target conditioned occlusion field to efficiently learn environment-aware affordance).
>
>
> > Section 4.3.1 is lacking a supporting Figure.
>
> Section 4.3.1 describes the Robot-target Conditioned Occlusion Field, and the 3 heatmaps in Figure 2 (Left), 3 heatmaps in Figure 3 show the most important area of the occlusion field. Additionally, the 8 heatmaps in Figure 6 in supplementary show the visualization of the whole occlusion fields, demonstrating how changing the target point and the robot position will influence the heatmap. Thank you for pointing out this! We will add more descriptions in Section 4.3.1 on the relationships between Section 4.3.1 and the above figures.
>
>
>
> > The robot geometry is not taken into account.
>
> Please see the **Geometry or configuration of the robot** section in **Global Response TEXT**.
>
>
>
>
> [1] Where2Act: From Pixels to Actions for Articulated 3D Objects. ICCV, 2021.
>
> [2] O2O-Afford: Annotation-Free Large-Scale Object-Object Affordance Learning. CoRL, 2021.
>
> [3] VAT-Mart: Learning Visual Action Trajectory Proposals for Manipulating 3D ARTiculated Objects. ICLR, 2022.
>
> [4] AdaAfford: Learning to Adapt Manipulation Affordance for 3D Articulated Objects via Few-shot Interactions. ECCV, 2022.
>
> [5] DualAfford: Learning Collaborative Visual Affordance for Dual-gripper Object Manipulation. ICLR, 2023.
>
> [6] PointNet++: Deep Hierarchical Feature Learning on Point Sets in a Metric Space. NeurIPS, 2017.

---

> > ### Comment · Reviewer_zyYG · 2023-08-14
> >
> > Thank you for the response. I think the paper would be valuable to share with the community and I do recommend accepting it.

---

> > > ### Author Response · Authors · 2023-08-21
> > > **Thank you!**
> > >
> > > We are glad that our responses help address your concerns. Thanks for your positive rating and recommendation to acceptance!

---

### Official Review · Reviewer_eVFQ · 2023-07-07

**Soundness:** 3 good
**Presentation:** 2 fair
**Contribution:** 3 good
**Rating:** 6
**Confidence:** 4

**Summary:**

The paper presents a novel environment-aware affordance prediction framework for robot interactions (pushing/pulling) with parts of articulated objects (drawers and doors). The proposed approach consists primarily of two novel improvements: a robot-target conditioned occlusion field and a contrastive learning formulation for improved data efficiency. The paper evaluates the proposed approach against where2act and O2O with metrics from those works and demonstrates a fair improvement (~6% overall and 2-10% in all individual categories). Real world trials demonstrate the usefulness of predicted point-level affordances for robotics tasks. Qualitative results demonstrate several clear wins for the approach related to prediction of occluding object constraints.

**Strengths:**

Based on a known simulation system and asset data, the evaluation and metrics soundly demonstrate the advantage of the proposed approach over prior art.
Trials on down-stream manipulation tasks with a real robot platform validate the usefulness of the point-level affordances predicted by the approach.
The proposed improvements over prior art (contrastive learning and robot-target conditioned occlusion fields) appear novel and constitute interesting solutions to the challenges posed.
The advantage of training on simple occlusions and generalizing to more complex occlusion scenarios is a clear strength of the approach.


**Weaknesses:**

Some areas of the paper could be improved. I felt there was some information missing leaving me with questions. The high-level concepts were repeated often and could be deduplicated to make room for more details.
The line was slightly blurry between novel work and prior art. Clearly this approach builds upon W2A and O2O, but the paper would benefit from clarifying carry-over vs. novel formulations.
	SMA metric appears to be the ssr metric from W2A with a different name.
	Some discussion of F-Score and Average Precision metrics would help interpretation. Could be appendix if repeated from prior art.
	Overall accuracy improvement over SoTA was somewhat small (6% over prior methods to achieve 43/36 push/pull).
	The paper talks up the approach without mentioning remaining challenges or failure modes.

Exposition could be improved with further polish and proofreading.
Grammar(G)/Typos(T) (listed with line number) :
G - 18 - perceiving and manipulating these objects present
T - 39 - also faces experiences
T - 107 - objectc
T - 148 - extra period after fR
G - 255 - We randomly adopt proposed
T - 267 - Moreover to we
274 - Table 3 2 (formatting? Comma or conjunction?)
G - 284 - of that may exist collision


**Questions:**

I understand the value of robot and target encoder networks, but I didn’t find any satisfying details or citations to provide insight into what these networks produce/encode or how they are trained. Are these empty MLPs initialized with random weights and allowed to fill freely, or is there something else to them? Are they described more in prior art or specific to this work?
I would like to better understand the intuition behind the Robot-Target Conditioned Occlusion Field. Why does the vector field of cross products of point-to-robot and point-to-target vectors provide a good heuristic? Some expanded text here would be valuable.
	What is a “novel” scene for the evaluation test metrics? Is it a scene with additional occluders that was not present in training?
Why does the pulling performance on novel scenes only improve 1-2%? Is this significantly harder to predict occlusions or is there a common failure mode that escapes the assumptions of the heuristic for the approach?


**Limitations:**

Discussion of limitations seemed light. Only the lack of mobile base was mentioned. Given the maximum performance of 43/36% push/pull I expected to learn more about what wasn’t working or what additional challenges and failure modes were observed.

---

> ### Author Rebuttal · Authors · 2023-08-10
>
> Thanks for your valuable comments! We have addressed them below. Hope to hear back from you if you have further questions!
>
> > High-level concepts were repeated often.
>
> Thanks for your suggestion! We will prune repetitive concepts to make space for crucial details.
>
>
> > Clarifications on carry-over vs. novel formulations.
>
> Given the overlap of many settings from W2A[1] and O2O[2] and limited pages, we didn't describe many details in the paper. We will further polish the paper and provide more details. Here are more clarifications.
> **Sample Manipulation Accuracy (SMA)** mirrors the Sample Successful Rate (SSR) in W2A. We think the name SMA can better reflect the function of affordance (providing actionable priors for manipulation), and thus use this name.
> For **F-Score and Average Precision**, we borrowed them from W2A and O2O. Our F1 Score calculation: $2 * (precision * recall) / (precision + recall)$. The definitions for precision and recall remain standard.
>
>
>
> > Challenges and failure modes.
>
> Please see **Global Response TEXT** and Figure D in the **Global Response PDF**.
>
>
> > Improvements over SoTA.
>
> **First**, baselines put up in the paper are not actual prior methods except for W2A. O2O, O2O-M, W2A-R have been modified by us to fit the new setting, which have been improved by adding consideration of robot constraints. Otherwise, prior works perform unsatisfactorily with a successful rate of 10%-23%. Besides, O2O-M is trained on data of scenes with multiple occluders, while our model is trained on data with only 1 occluder.
> **Second**, as described in *Challenges and failure modes*, 80% failures come from the initial close distances between the robot and some occluders, or the following motions of some robot joints given a plausible target point. These cases are quite difficult and require a policy to produce fine-grained robot joint variable sequences equipped with our environment-aware actionable priors that propose plausible target points. We recognize this as a prospective avenue for future research. Our environment-aware affordance serves as a pioneering step in this direction, highlighting the significance of identifying credible target points.When excluding such failure modes, our method performs well in most cases and the improvement margin over baselines becomes larger.
>
>
>
> > Robot and target encoder networks.
>
> The robot encoder E_R is a 3-layer MLP (3->32->64->128) to represent the robot’s 3D position relative to target objects. The target encoder is a Segmentation-version PointNet++[3] encoder (architecture same as [3]). Following prior works [1, 2], all networks are initialized with random weights and allowed to fill freely.
>
> > Why Robot-Target Conditioned Occlusion Field provides a good heuristic.
>
> The intuition of Robot-Target Conditioned Occlusion Field is to highlight points near the robot or target, while sidelining distant ones. Therefore, we design the vector field in a manner that represents the spatial relationships of points in the scene *relative to both the robot and the target*.
>
> By defining the field factors $V_R$ and $V_{T_p}$ as vectors pointing from any scene point to the robot and the target respectively, the vectors inherently encode the spatial relation of the scene points to the robot and the target. The cross product of two vectors provides a new vector that is orthogonal to the plane formed by the original vectors. By taking the cross product of $V_R$ and $V_{T_p}$, the resulting field captures the interaction between a point's relationship with the robot and the target.
> It emphasizes points that have significant relationships with both the robot and the target.
> The field value approaches zero as a point gets closer to robot/target. This property ensures the points that have a large cross product of $V_R$ and $V_{T_p}$ can be considered less significant since their spatial relationship to both the robot and target is more distant or oblique while keeping near important points.
>
> Moreover, **Figure 6 in supplementary** visualizes many occlusion fields showing the influence on field values caused by different robot positions / target points.
>
> > “Novel” scenes for the evaluation test metrics.
>
> Novel refers to **unseen object categories**. Our model is trained on objects in 8 categories, tested on combinations of objects in 4 unseen categories. Details and statistics are shown in Table 1 in the paper. Thanks for this comment, and we will further clarify this setting.
>
>
> > Pulling performance on novel scenes.
>
> Such performance improves 1-2% only for **F-Score metric compared with W2A**, and **Average Precision metric compared with W2A-R and O2O**. For **W2A**, as it only considers a flying gripper without considering the constraints of the robot arm, more points are predicted as actionable (positive), which leads to the recall (true positive rate) super higher than other methods, leading F-score to be high, while with a much lower Average Precision (false positive rate) than all other methods. For **W2A-R and O2O**, as most points are not pullable, models tend to give negative predictions, and only actions that are more likely to succeed will be predicted as positive by these models and ours. As Average Precision is calculated among actions predicted as positive, the performance of these models and ours in this metric will be similar in the pulling task. With similar precision, as our method has a higher recall than W2A-R and O2O, it has a much higher F-Score. Overall our model is robust and performs well in 4 tasks as a whole.
>
> > Grammar / Typos and improvement on exposition.
>
> Thanks for your attention to details! We will fix the problems and carefully improve the exposition.
>
>
>
> [1] Where2Act: From Pixels to Actions for Articulated 3D Objects. ICCV, 2021.
>
> [2] O2O-Afford: Annotation-Free Large-Scale Object-Object Affordance Learning. CoRL, 2021.
>
> [3] PointNet++: Deep Hierarchical Feature Learning on Point Sets in a Metric Space. NeurIPS, 2017.

---

> > ### Comment · Reviewer_eVFQ · 2023-08-16
> >
> > I first want to thank the authors for their detailed responses to my questions and concerns. Given the content of the rebuttal and assuming that the final text will include the recommended improvements in the detail and clarity across the reviews, I've increase my positive rating and feel this work should be accepted.
> >
> > I do think there is ample room in the paper to add content as repetitive high-level concepts are pruned and de-duplicated. I encourage the authors to provide more intuition, add concrete implementation/architecture details, and clarify the distinction between this work and prior art.

---

> > > ### Author Response · Authors · 2023-08-21
> > > **Thank you!**
> > >
> > > Thank you for increasing your positive rating! We are glad that our responses help address your concerns, and will include the recommended improvements in the final text.

---

### Official Review · Reviewer_gQVm · 2023-07-07

**Soundness:** 3 good
**Presentation:** 3 good
**Contribution:** 3 good
**Rating:** 5
**Confidence:** 4

**Summary:**

This paper introduces a framework for learning environment-aware affordance in occluded 3D object manipulation. It tackles combinatorial explosion and improves data efficiency. The authors showcase their superior framework using benchmarking multi-object full-robot environments in the SAPIEN simulator. Their contributions involve studying articulated object manipulation within environment constraints, proposing a data-efficient framework, and establishing benchmarking environments.

**Strengths:**

- The framework proposed addresses the learning of environment-aware affordances for 3D articulated object manipulation, even under occlusions.
- The paper focuses on affordance learning and introduces benchmarking environments to enhance robotic system performance in real-world settings.
- The paper includes a comprehensive description of the framework, establishes benchmarking environments, and highlights its superior performance.
- It is well-structured, easy to comprehend, and effectively presents experimental results.


**Weaknesses:**

- The proposed framework has only been evaluated on simulated environments, and there is a lack of real-world experiments from different perspectives and scenarios.
- This work primarily examines a static scene, which raises concerns about the applicability of the proposed framework in real-world scenarios that involve more complex and dynamic environments.
- The authors have not provided a detailed analysis of the failure cases of the proposed framework.


**Questions:**

- L38/267 mention that sampling-based motion planning faces performance degradation with increasing occluders. Could you provide a benchmark of the time and cost of different methods to address this issue?
-  L147 mention E_R encodes the robot position R into a latent representation. What about the geometry or configuration of the robot? How do you ensure that the scene and the configuration of the robot are plausible?
- How would the proposed framework generalize to different types of robots and scenes?
- In real-time executions, the state of point clouds is constantly changing, and the heatmap also changes. How do you plan to handle this issue?

**Limitations:**

The authors have addressed the limitations and potential negative societal impact of their work.

---

> ### Author Rebuttal · Authors · 2023-08-10
>
> Thanks for your valuable comments. We have addressed them below, and hope to hear back from you if you have further questions!
>
> > Lack of real-world exp from different perspectives and scenarios.
>
> We have provided **3 pairs of comparisons of affordance and their corresponding manipulation policies in real-world scenarios** in Figure 7 in the supplementary pdf and the supplementary video (3:35 - 4:09), to show the superiority of our method over baselines in the real world. Additionally, we have scanned **more real-world point clouds in different perspectives and scenarios** and provided the predicted affordance in Figure C in the Global Response PDF. Since point clouds of different perspectives can be rotated into the same perspective with differences in the vacancy of point clouds given camera intrinsics and extrinsics, our method is robust to the complexity of perspectives (except for orthogonal and behind views). Besides, apart from our work, a series of papers have demonstrated point-level affordance can generate promising results when directly deployed on different real-world point scans [1, 3, 4, 5] (specifically, Figure 6 in [1], Figure 5 in [3], Figure 6 in [4], Figure 1, 7 in [5] and Figure 8 in supplementary in [4]).
>
>
>
> > The applicability of the proposed framework in real-world scenarios that involve more complex and dynamic environments.
>
> Following the settings of previous affordance learning works [1, 2, 3, 4, 5], we evaluate our framework in static environments. We have made the scene more complex and realistic by introducing **combinations of occluders** in the environment, and then integrate object-centric actionable priors with environment constraints. As for **manipulation in a dynamic scene**, a simple solution is to manipulate step-by-step with the affordance of each frame, which is demonstrated in [6, 7]. As shown in Figure E in the **Global Response PDF**, we construct a dynamic scene with the movement of the target articulated part and one occluder. The predicted affordance in such a dynamic scene shows **consistency** in consecutive frames as most points do not change much. We believe making affordance more consistent in such dynamic scenes is a future direction worth studying. As for real-time dynamic scenes with movable robots, we believe it is a promising direction for future work, and our work has laid a solid foundation in this direction. Thanks for this valuable comment! We will add more results and discussions in the paper.
>
> > Changing states of point clouds and heatmaps in real-time executions.
>
> Please refer to the previous response.
>
>
>
> > Detailed analysis of failure cases.
>
> See the **Global Response TEXT** and Figure D in the **Global Response PDF** for details.
>
>
> > Benchmark of the time and cost of different methods on the issue of performance degradation with increasing occluders.
>
> Performance degradation with increasing occluders is a common issue for planning, as it costs more to find a successful path. In contrast, the time cost of our method remains constant, because the complexity of different input point clouds does not affect the inference time of neural networks at a fixed point sampling number. Below we provide the benchmark of time and cost of different methods: the commonly used sampling-based method RRT for planning and our method. For RRT, when the number of occlusions increases from 0 to 5, the inference time in one scene is 0.04, 0.27, 1.38, 2.29, 2.75, 3.08 seconds respectively, while the inference time of our method remains around 0.18 seconds.
>
>
>
> > Geometry or configuration of the robot.
>
> Please see the **Geometry or configuration of the robot** section in **Global Response TEXT**.
>
>
>
> > Generalization to different types of robots and scenes.
>
> For different robot types, please see the **Geometry or configuration of the robot** section in **Global Response TEXT**. For different scenes, our environment-aware affordance has **demonstrated good generalization** towards scenes different from those in training. Specifically, a scene is mainly composed of the target object and occluders. For **target objects**, point-level affordance has demonstrated the generalization towards different target objects [1, 2, 3, 4, 5], because it focuses on the local geometry of each target point instead of the whole object, which is more generalizable across different object categories sharing similar actionable local parts. Apart from the two categories (Cabinet and Table) in our paper, we further demonstrate the affordance of **more categories** in Figure B in the **Global Response PDF**. For **occluders**, our proposed Robot-Target Occlusion Field filters out most unimportant points and focuses on the most important part of the scene, facilitating generalization to novel scenes with unseen and more occluders. As a result, while trained on scenes with **only one occluder** sampled from 8 kinds of 367 objects, our model can generalize to **combinations** of 128 **unseen objects** in training categories and 589 objects in 4 **unseen categories** in a scene. Statistics and example occluders are shown in Table 1 in the main paper and Figure 1 in the supplementary PDF.
>
> [1] Where2Act: From Pixels to Actions for Articulated 3D Objects. ICCV, 2021.
>
> [2] O2O-Afford: Annotation-Free Large-Scale Object-Object Affordance Learning. CoRL, 2021.
>
> [3] VAT-Mart: Learning Visual Action Trajectory Proposals for Manipulating 3D ARTiculated Objects. ICLR, 2022.
>
> [4] AdaAfford: Learning to Adapt Manipulation Affordance for 3D Articulated Objects via Few-shot Interactions. ECCV, 2022.
>
> [5] DualAfford: Learning Collaborative Visual Affordance for Dual-gripper Object Manipulation. ICLR, 2023.
>
> [6] UMPNet: Universal Manipulation Policy Network for Articulated Objects. ICRA, 2022.
>
> [7] FlowBot3D: Learning 3D Articulation Flow to Manipulate Articulated Objects. RSS, 2022.

---

> > ### Comment · Reviewer_gQVm · 2023-08-16
> >
> > Thank you for your detailed responses and clarifications. You have effectively addressed my concerns, and I maintain my positive rating.

---

> > > ### Author Response · Authors · 2023-08-21
> > > **Thank you!**
> > >
> > > We are glad that our responses help address your concerns. Thanks for your positive rating!

---

### Author Rebuttal · Authors · 2023-08-10

We extend our gratitude to reviewers for their careful reading, meticulous feedback and valuable insights.
We are glad that reviewers unanimously agree that our work (1) proposes meaningful environment-aware affordance (zyYG: “a step up from previous work”, ntD4: “novel and insightful”, wNB2: “address several important limitations”), (2) contains many insightful designs (eVFQ:  “novel and interesting”, zyYG: “several novelties”, ntD4: “intriguing”) and (3) shows comprehensive experiments (eVFQ: “soundly demonstrate advantage”, zyYG: “good analysis”, wnB2: “adequate”).

In this **Global Response TEXT**, we provide clarifications to 3 common concerns.


> Geometry or configuration of the robot. (Reviewer gQVm&zyYG)

We clarify this concern by respectively clarifying (1.1) using 1 robot type is a realistic setting that previous works employ; (1.2) in this setting, the training data without the robot point cloud implicitly reflects robot geometry/configuration; (2) our framework can accommodate scenarios with multiple robot types when encoding robot point clouds.

For (1.1), contrary to various targets, robot types are generally limited, and thus existing robotic manipulation works [1, 3, 4, 5] and real-world scenarios usually use 1 type of robot/gripper. We follow them and use Franka Panda.

For (1.2), in this single-robot-type setting, affordance can be learned without inputting robot point clouds. Given that only one robot type features across scenes, both its initial configuration and point cloud (geometry) remain constant, and the only difference comes from the robot's position R. So R is enough to represent robots’ difference across different scenes. The interaction results between the robot and different targets can reflect its geometry and configuration. Specifically, supervised by actionable information of robot-object interaction, the learned affordance includes the robot’s geometry and configuration from interaction results implicitly. For this reason, previous manipulation works [1, 3, 4, 5] learn the manipulation affordance/policy without encoding the robot point cloud.

For (2), when existing many types of robot, we have extended our model to take the robot point cloud as extra input, on 3 different types (xArm7, Jaco2 and Franka Panda). Illustrated in Figure A of **Global Response PDF**, our learned affordance adeptly mirrors robots’ configuration and geometry. We provide a detailed analysis of affordance visualization in the figure caption.

Such **ability of PointNet++ encoders to reflect geometry of two interacting objects** is also proven in O2OAfford [2] which studies affordance between objects.



> Challenges and failure modes. (Reviewer gQVm&eVFQ)

After analyzing the visualization gallery of our failed cases, our failures basically follows 2 modes:

(1) The starting position of the robot is closely adjacent to some occluders, and the robot initially collides with occlusions even in small-scale movement. **This case is demonstrated in the first row of Figure D in the Global Response PDF**. This mode accounts for about *55%* of the failure cases.

(2) Though the environment-aware affordance model proposes a plausible target point, the following actions of robot joints lead to collisions. Given the target point (i.e., the final position of the gripper), we employ inverse kinematics (IK), a prevalent method in robotics to calculate the robot joint configurations that would result in a desired end-effector pose.
However, IK can produce varied joint configurations that lead to the same endpoint for the gripper. Some samples may cause a robot link to have large rotations/translations when approaching target. Consequently, these links risk colliding with occluders. **This case is demonstrated in the second row of Figure D in the Global Response PDF**. This mode accounts for about *25%* of the failure cases. Note that, even though IK is the most commonly used for solving the joint variables and works well with RRT in non-occlusion scenarios, we believe in our novel difficult setting with multiple occluders, the joint motions should be better planned. As all baselines use IK for moving joints, it is clear that our proposed environment-aware affordance has already proposed more plausible points as potential targets.


These 2 modes together account for about *80%* of failure cases. Equipped with our environment-aware actionable priors to propose plausible target manipulation points, these cases further require a policy to produce fine-grained action sequences of each robot joint. We recognize this as a prospective avenue for future research. Our environment-aware affordance serves as a pioneering step in this direction, highlighting the significance of identifying credible target points.



> Segmentation mask. (Reviewer ntD4&wNB2)

As our framework needn’t consider each occluder one by one, we only need 1 mask of the target object, and the points left are on occluders. This setting is similar to previous affordance learning works [1, 3, 4] that require the mask of target parts.
In simulation, the mask can be acquired by the simulator. For real cases, we can easily acquire the mask of the target object using segmentation models like SAM [6].


We appreciate the valuable and insightful comments from reviewers! We will add more details and discussions in the paper.

[1] Where2Act: From Pixels to Actions for Articulated 3D Objects. ICCV, 2021.

[2] O2O-Afford: Annotation-Free Large-Scale Object-Object Affordance Learning. CoRL, 2021.

[3] VAT-Mart: Learning Visual Action Trajectory Proposals for Manipulating 3D ARTiculated Objects. ICLR, 2022.

[4] AdaAfford: Learning to Adapt Manipulation Affordance for 3D Articulated Objects via Few-shot Interactions. ECCV, 2022.

[5] UMPNet: Universal Manipulation Policy Network for Articulated Objects. ICRA, 2022.

[6] Segment Anything. ArXiv, 2023.

---

### Decision · Program_Chairs · 2023-09-21

**Decision:**

Accept (poster)

**Comment:**

This paper received positive reviewer opinions after the discussion period.  The main concerns expressed during the discussion revolved around more clearly explaining in the writeup some important elements: the problem definition, technical details of the encoder network architecture, and delineating the contributions of this work relative to prior work.  The author responses during the disucssion alleviated these concerns.  The AC finds no basis to overturn the reviewer consensus and therefore recommends acceptance.  The AC strongly recommends a thorough editing pass to incorporate the clarifications and improvements from this discussion period when preparing the camera ready version.